# DECOrrelated feature space partitioning for distributed sparse regression

**Xiangyu Wang**
Dept. of Statistical Science
Duke University
wwrechard@gmail.com

**David Dunson**
Dept. of Statistical Science
Duke University
dunson@stat.duke.edu

**Chenlei Leng**
Dept. of Statistics
University of Warwick
C.Leng@warwick.ac.uk

## Abstract

Fitting statistical models is computationally challenging when the sample size or the dimension of the dataset is huge. An attractive approach for down-scaling the problem size is to first partition the dataset into subsets and then fit using distributed algorithms. The dataset can be partitioned either horizontally (in the sample space) or vertically (in the feature space). While the majority of the literature focuses on sample space partitioning, feature space partitioning is more effective when $p \gg n$. Existing methods for partitioning features, however, are either vulnerable to high correlations or inefficient in reducing the model dimension. In this paper, we solve these problems through a new embarrassingly parallel framework named DECO for distributed variable selection and parameter estimation. In DECO, variables are first partitioned and allocated to $m$ distributed workers. The decorrelated subset data within each worker are then fitted via any algorithm designed for high-dimensional problems. We show that by incorporating the decorrelation step, DECO can achieve consistent variable selection and parameter estimation on each subset with (almost) no assumptions. In addition, the convergence rate is nearly minimax optimal for both sparse and weakly sparse models and does NOT depend on the partition number $m$. Extensive numerical experiments are provided to illustrate the performance of the new framework.

## 1 Introduction

In modern science and technology applications, it has become routine to collect complex datasets with a huge number $p$ of variables and/or enormous sample size $n$. Most of the emphasis in the literature has been on addressing large $n$ problems, with a common strategy relying on partitioning data samples into subsets and fitting a model containing all the variables to each subset [1, 2, 3, 4, 5, 6]. In scientific applications, it is much more common to have huge $p$ small $n$ data sets. In such cases, a sensible strategy is to break the features into groups, fit a model separately to each group, and combine the results. We refer to this strategy as feature space partitioning, and to the large $n$ strategy as sample space partitioning.

There are several recent attempts on parallel variable selection by partitioning the feature space. [7] proposed a Bayesian split-and-merge (SAM) approach in which variables are first partitioned into subsets and then screened over each subset. A variable selection procedure is then performed on the variables that survive for selecting the final model. One caveat for this approach is that the algorithm cannot guarantee the efficiency of screening, i.e., the screening step taken on each subset might select a large number of unimportant but correlated variables [7], so SAM could be ineffective in reducing the model dimension. Inspired by a group test, [8] proposed a parallel feature selection algorithm by repeatedly fitting partial models on a set of re-sampled features, and then aggregating the residuals to

form scores for each feature. This approach is generic and efficient, but the performance relies on a strong condition that is almost equivalent to an independence assumption on the design.

Intuitively, feature space partitioning is much more challenging than sample space partitioning, mainly because of the correlations between features. A partition of the feature space would succeed only when the features across the partitioned subsets were mutually independent. Otherwise, it is highly likely that any model posed on the subsets is mis-specified and the results are biased regardless of the sample size. In reality, however, mutually independent groups of features may not exist; Even if they do, finding these groups is likely more challenging than fitting a high-dimensional model. Therefore, although conceptually attractive, feature space partitioning is extremely challenging.

On the other hand, feature space partitioning is straightforward if the features are independent. Motivated by this key fact, we propose a novel embarrassingly-parallel framework named DECO by *decorrelating* the features before partitioning. With the aid of decorrelation, each subset of data after feature partitioning can now produce consistent estimates even though the model on each subset is intrinsically mis-specified due to missing features. To the best of our knowledge, DECO is the first embarrassingly parallel framework accommodating arbitrary correlation structures in the features. We show, quite surprisingly, that the DECO estimate, by leveraging the estimates from subsets, achieves the same convergence rate in $\ell_2$ norm and $\ell_\infty$ norm as the estimate obtained by using the full dataset, and that the rate does not depend on the number of partitions. In view of the huge computational gain and the easy implementation, DECO is extremely attractive for fitting large-$p$ data.

The most related work to DECO is [9], where a similar procedure was introduced to improve *lasso*. Our work differs substantially in various aspects. First, our motivation is to develop a parallel computing framework for fitting large-$p$ data by splitting features, which can potentially accommodate any penalized regression methods, while [9] aim solely at complying with the irrepresentable condition for *lasso*. Second, the conditions posed on the feature matrix are more flexible in DECO, and our theory, applicable for not only sparse signals but also those in $l_r$ balls, can be readily applied to the preconditioned *lasso* in [9].

The rest of the paper is organized as follows. In Section 2, we detail the proposed framework. Section 3 provides the theory of DECO. In particular, we show that DECO is consistent for both sparse and weakly sparse models. Section 4 presents extensive simulation studies to illustrate the performance of our framework. In Section 5, we outline future challenges and future work. All the technical details are relegated to the Appendix.

## 2    Motivation and the DECO framework

Consider the linear regression model
$$Y = X\beta + \varepsilon, \tag{1}$$
where $X$ is an $n \times p$ feature (design) matrix, $\varepsilon$ consists of $n$ *i.i.d* random errors and $Y$ is the response vector. A large class of approaches estimate $\beta$ by solving the following optimization problem

$$\hat{\beta} = arg \min_{\beta} \frac{1}{n} \|Y - X\beta\|_2^2 + 2\lambda_n \rho(\beta),$$

where $\| \cdot \|_2$ is the $\ell_2$ norm and $\rho(\beta)$ is a penalty function. In this paper, we specialize our discussion to the $\ell_1$ penalty where $\rho(\beta) = \sum_{j=1}^{p} |\beta_j|$ [10] to highlight the main message of the paper.

As discussed in the introduction, a naive partition of the feature space will usually give unsatisfactory results under a parallel computing framework. That is why a decorrelation step is introduced. For data with $p \leq n$, the most intuitive way is to orthogonalize features via the singular value decomposition (SVD) of the design matrix as $X = UDV^T$, where $U$ is an $n \times p$ matrix, $D$ is an $p \times p$ diagonal matrix and $V$ an $p \times p$ orthogonal matrix. If we pre-multiply both sides of (1) by $\sqrt{p}UD^{-1}U^T = (XX^T/p)^{\frac{+}{2}}$, where $A^+$ denotes the Moore-Penrose pseudo-inverse, we get

$$\underbrace{(XX^T/p)^{\frac{+}{2}}Y}_{\tilde{Y}} = \underbrace{\sqrt{p}UV^T}_{\tilde{X}}\beta + \underbrace{(XX^T/p)^{\frac{+}{2}}\varepsilon}_{\tilde{\varepsilon}}. \tag{2}$$

It is obvious that the new features (the columns of $\sqrt{p}UV^T$) are mutually orthogonal. Define the new data as $(\tilde{Y}, \tilde{X})$. The mutually orthogonal property allows us to decompose $\tilde{X}$ column-wisely to

$m$ subsets $\tilde{X}^{(i)}, i = 1, 2, \cdots, m$, and still retain consistency if one fits a linear regression on each subset. To see this, notice that each sub-model now takes a form of $\tilde{Y} = \tilde{X}^{(i)}\beta^{(i)} + \tilde{W}^{(i)}$ where $\tilde{W}^{(i)} = \tilde{X}^{(-i)}\beta^{(-i)} + \tilde{\varepsilon}$ and $X^{(-i)}$ stands for variables not included in the $i^{th}$ subset. If, for example, we would like to compute the ordinary least squares estimates, it follows

$$\hat{\beta}^{(i)} = (\tilde{X}^{(i)T}\tilde{X}^{(i)})^{-1}\tilde{X}^{(i)}\tilde{Y} = \beta^{(i)} + (\tilde{X}^{(i)T}\tilde{X}^{(i)})^{-1}\tilde{X}^{(i)}\tilde{W}^{(i)} = \beta^{(i)} + (\tilde{X}^{(i)T}\tilde{X}^{(i)})^{-1}\tilde{X}^{(i)}\tilde{\varepsilon},$$

where $\hat{\beta}^{(i)}$ converges at the same rate as if the full dataset were used.

When $p$ is larger than $n$, the new features are no longer exactly orthogonal to each other due to the high dimension. Nevertheless, as proved later in the article, the correlations between different columns are roughly of the order $\sqrt{\log p/n}$ for random designs, making the new features approximately orthogonal when $\log(p) \ll n$. This allows us to follow the same strategy of partitioning the feature space as in the low dimensional case. It is worth noting that when $p > n$, the SVD decomposition on $X$ induces a different form on the three matrices, i.e., $U$ is now an $n \times n$ orthogonal matrix, $D$ is an $n \times n$ diagonal matrix, $V$ is an $n \times p$ matrix, and $\left(\frac{XX^T}{p}\right)^{\frac{+}{2}}$ becomes $\left(\frac{XX^T}{p}\right)^{-\frac{1}{2}}$.

In this paper, we primarily focus on datasets where $p$ is so large that a single computer is only able to store and perform operations on an $n \times q$ matrix ($n < q < p$) but not on an $n \times p$ matrix. Because the two decorrelation matrices yield almost the same properties, we will only present the algorithm and the theoretical analysis for $(XX^T/p)^{-1/2}$.

The concrete DECO framework consists of two main steps. Assume $X$ has been partitioned column-wisely into $m$ subsets $X^{(i)}, i = 1, 2, \cdots, m$ (each with a maximum of $q$ columns) and distributed onto $m$ machines with $Y$. In the first stage, we obtain the decorrelation matrix $(XX^T/p)^{-1/2}$ or $\sqrt{p}D^{-1}U^T$ by computing $XX^T$ in a distributed way as $XX^T = \sum_{i=1}^{m} X^{(i)}X^{(i)T}$ and perform the SVD decomposition on $XX^T$ on a central machine. In the second stage, each worker receives the decorrelation matrix, multiplies it to the local data $(Y, X^{(i)})$ to obtain $(\tilde{Y}, \tilde{X}^{(i)})$, and fits a penalized regression. When the model is assumed to be exactly sparse, we can potentially apply a refinement step by re-estimating coefficients on all the selected variables simultaneously on the master machine via ridge regression. The details are provided in Algorithm 1.

The entire Algorithm 1 contains only two map-reduce passes and is thus communication-efficient. Lines 14 - 18 in Algorithm 1 are added only for the data analysis in Section 5.3, in which $p$ is massive compared to $n$ in that $\log(p)$ is comparable to $n$, and the algorithm may not scale down the size of $p$ sufficiently for even obtaining a ridge regression estimator afterwards. Thus, a further sparsification step is recommended. The condition in Line 16 is only triggered in our last experiment, but is crucial for improving the performance for extreme cases. In Line 5, the algorithm inverts $XX^T + r_1 I$ instead of $XX^T$ for robustness, because the rank of $XX^T$ after standardization will be $n - 1$. Using ridge refinement instead of ordinary least squares is also for robustness. The precise choice of $r_1$ and $r_2$ will be discussed in the numerical section.

Penalized regression fitted using regularization path usually involves a computational complexity of $\mathcal{O}(knp + kd^2)$, where $k$ is the number of path segmentations and $d$ is the number of features selected. Although the segmentation number $k$ could be as bad as $(3^p + 1)/2$ in the worst case [11], real data experience suggests that $k$ is on average $\mathcal{O}(n)$ [12], thus the complexity for DECO takes a form of $\mathcal{O}\left(n^3 + n^2\frac{p}{m} + m\right)$ in contrast to the full *lasso* which takes a form of $\mathcal{O}(n^2p)$.

## 3 Theory

In this section, we provide theoretical justification for DECO on random feature matrices. We specialize our attention to *lasso* due to page limits and will provide the theory on general penalties in the long version. We prove the consistency results for the estimator obtained after Stage 2 of DECO, while the consistency of Stage 3 will then follow immediately. For simplicity, we assume that $\varepsilon$ follows a sub-Gaussian distribution and $X \sim N(0, \Sigma)$ throughout this section, although the theory can be easily extended to the situation where $X$ follows an elliptical distribution and $\varepsilon$ is heavy-tailed.

Recall that DECO fits the following linear regression on each worker

$$\tilde{Y} = \tilde{X}^{(i)}\beta^{(i)} + \tilde{W}^{(i)}, \quad \text{and} \quad \tilde{W}^{(i)} = \tilde{X}^{(-i)}\beta^{(-i)} + \tilde{\varepsilon},$$

---

**Algorithm 1** *The DECO framework*

---

**Initialization:**
 1: Input $(Y, X), p, n, m, \lambda_n$. Standardize $X$ and $Y$ to $x$ and $y$ with mean zero;
 2: Partition (arbitrarily) $(y, x)$ into $m$ disjoint subsets $(y, x^{(i)})$ and distribute to $m$ machines;
**Stage 1 : Decorrelation**
 3: On each worker, compute $x^{(i)} x^{(i)T}$ and push to the center machine;
 4: On the center machine, compute $F = \sum_{i=1}^{m} x^{(i)} x^{(i)T}$;
 5: $\bar{F} = \sqrt{p} (F + r_1 I_p)^{-1/2}$;
 6: Push $\bar{F}$ to each worker.
 7: **for** $i = 1$ **to** $m$ **do**
 8: $\quad \tilde{y} = \bar{F} y$ and $\tilde{x}^{(i)} = \bar{F} x^{(i)}$; # obtain decorrelated data
 9: **end for**
**Stage 2 : Estimation**
10: On each worker we estimate $\hat{\beta}^{(i)} = arg \min_{\beta} \frac{1}{n} \|\tilde{y} - \tilde{x}^{(i)} \beta\|_2^2 + 2\lambda_n \rho(\beta)$;
11: Push $\hat{\beta}^{(i)}$ to the center machine and combine $\hat{\beta} = (\hat{\beta}^{(1)}, \hat{\beta}^{(2)}, \cdots, \hat{\beta}^{(m)})$;
12: $\hat{\beta}_0 = mean(Y) - mean(X)^T \hat{\beta}$ for intercept.
**Stage 3 : Refinement (optional)**
13: **if** $\#\{\hat{\beta} \neq 0\} \geq n$ **then**
14: $\quad$ # *S*parsification is needed before ridge regression.
15: $\quad \mathcal{M} = \{k : |\hat{\beta}_k| \neq 0\}$;
16: $\quad \hat{\beta}_{\mathcal{M}} = arg \min_{\beta} \frac{1}{n} \|\tilde{y} - \tilde{x}_{\mathcal{M}} \beta\|_2^2 + 2\lambda_n \rho(\beta)$;
17: **end if**
18: $\mathcal{M} = \{k : |\hat{\beta}_k| \neq 0\}$;
19: $\hat{\beta}_{\mathcal{M}} = (X_{\mathcal{M}}^T X_{\mathcal{M}} + r_2 I_{|\mathcal{M}|})^{-1} X_{\mathcal{M}}^T Y$;
20: Return $\hat{\beta}$;

---

where $X^{(-i)}$ stands for variables not included in the $i^{th}$ subset. Our proof relies on verifying each pair of $(\tilde{Y}, \tilde{X}^{(i)}), i = 1, 2, \cdots, m$ satisfies the consistency condition of lasso for the random features. Due to the page limit, we only state the main theorem in the article and defer all the proofs to the supplementary materials.

**Theorem 1** (s-sparse). *Assume that $\beta_*$ is an s-sparse vector. Define $\sigma_0^2 = var(Y)$. For any $A > 0$ we choose $\lambda_n = A\sigma_0 \sqrt{\frac{\log p}{n}}$. Now if $p > c_0 n$ for some $c_0 > 1$ and $64 C_0^2 A^2 s^2 \frac{\log p}{n} \leq 1$, then with probability at least $1 - 8p^{1 - C_1 A^2} - 18pe^{-Cn}$ we have*

$$\|\hat{\beta} - \beta_*\|_\infty \leq \frac{5 C_0 A \sigma_0}{8} \sqrt{\frac{\log p}{n}} \quad and \quad \|\hat{\beta} - \beta_*\|_2^2 \leq \frac{9 C_0 A^2 \sigma_0^2}{8} \frac{s \log p}{n},$$

*where $C_0 = \frac{8c^*}{c_1 c_*}$ and $C_1 = \min\{\frac{c_* c_0^2}{8c^* c_2 (1 - c_0)^2}, \frac{c_*^3}{8 c_4^2 c^{*2}}\}$ are two constants and $c_1, c_2, c_4, c_*, c^*, C$ are defined in Lemma 6 in the supplementary materials. Furthermore, if we have*

$$\min_{\beta_k \neq 0} |\beta_k| \geq \frac{C_0 A \sigma_0}{4} \sqrt{\frac{\log p}{n}},$$

*then $\hat{\beta}$ is **sign consistent**, i.e., $sign(\hat{\beta}_k) = sign(\beta_k), \forall \beta_k \neq 0$ and $\hat{\beta}_k = 0, \forall \beta_k = 0$.*

Theorem 1 looks a bit surprising since the convergence rate does not depend on $m$. This is mainly because the bounds used to verify the consistency conditions for lasso hold uniformly on all subsets of variables. For subsets where no true signals are allocated, *lasso* will estimate all coefficients to be zero, so that the loss on these subsets will be exactly zero. Thus, when summing over all subsets, we retrieve the $\frac{s \log p}{n}$ rate. In addition, it is worth noting that Theorem 1 guarantees the $\ell_\infty$ convergence and sign consistency for *lasso* without assuming the irrepresentable condition [13]. A similar but weaker result was obtained in [9].

**Theorem 2** ($l_r$-ball). *Assume that $\beta_* \in \mathbb{B}(r, R)$ and all conditions in Theorem 1 except that $64 C_0^2 A^2 s^2 \frac{\log p}{n} \leq 1$ are now replaced by $64 C_0^2 A^2 R^2 \left(\frac{\log p}{n}\right)^{1-r} \leq 1$. Then with probability at least*

$1 - 8p^{1-C_1A^2} - 18pe^{-Cn}$, *we have*

$$\|\hat{\beta} - \beta_*\|_\infty \leq \frac{3C_0A\sigma_0}{2}\sqrt{\frac{\log p}{n}} \quad and \quad \|\hat{\beta} - \beta_*\|_2^2 \leq \left(\frac{9C_0}{8} + 38\right)(A\sigma_0)^{2-r}R\left(\frac{\log p}{n}\right)^{1-\frac{r}{2}}.$$

Note that $\sigma_0^2 = var(Y)$ instead of $\sigma$ appears in the convergence rate in both Theorem 1 and 2, which is inevitable due to the nonzero signals contained in $\tilde{W}$. Compared to the estimation risk using full data, the results in Theorem 1 and 2 are similar up to a factor of $\sigma^2/\sigma_0^2 = 1 - \hat{R}^2$, where $\hat{R}^2$ is the coefficient of determination. Thus, for a model with an $\hat{R}^2 = 0.8$, the risk of DECO is upper bounded by five times the risk of the full data inference. The rates in Theorem 1 and 2 are nearly minimax-optimal [14, 15], but the sample requirement $n \asymp s^2$ is slightly off the optimal. This requirement is rooted in the $\ell_\infty$-convergence and sign consistency and is almost unimprovable for random designs. We will detail this argument in the long version of the paper.

## 4 Experiments

In this section, we present the empirical performance of DECO via extensive numerical experiments. In particular, we compare DECO after 2 stage fitting (**DECO-2**) and DECO after 3 stage fitting (**DECO-3**) with the full data *lasso* (**lasso-full**), the full data *lasso* with ridge refinement (**lasso-refine**) and *lasso* with a naive feature partition without decorrelation (**lasso-naive**). This section consists of three parts. In the first part, we run **DECO-2** on some simulated data and monitor its performance on one randomly chosen subset that contains part of the true signals. In the second part, we verify our claim in Theorem 1 and 2 that the accuracy of DECO does not depend on the subset number. In the last part, we provide a comprehensive evaluation of DECO's performance by comparing DECO with other methods under various correlation structures.

The synthetic datasets are from model (1) with $X \sim N(0, \Sigma)$ and $\varepsilon \sim N(0, \sigma^2)$. The variance $\sigma^2$ is chosen such that $\hat{R}^2 = var(X\beta)/var(Y) = 0.9$. We consider five different structures of $\Sigma$.

**Model (i)** *Independent predictors*. The support of $\beta$ is $S = \{1, 2, 3, 4, 5\}$. We generate $X_i$ from a standard multivariate normal distribution with independent components. The coefficients are specified as

$$\beta_i = \begin{cases} (-1)^{Ber(0.5)}\left(|N(0,1)| + 5\sqrt{\frac{\log p}{n}}\right) & i \in S \\ 0 & i \notin S. \end{cases}$$

**Model (ii)** *Compound symmetry*. All predictors are equally correlated with correlation $\rho = 0.6$. The coefficients are the same as those in Model (i).

**Model (iii)** *Group structure*. This example is Example 4 in [16], for which we allocate the 15 true variables into three groups. Specifically, the predictors are generated as $x_{1+3m} = z_1 + N(0, 0.01)$, $x_{2+3m} = z_2 + N(0, 0.01)$ and $x_{3+3m} = z_3 + N(0, 0.01)$, where $m = 0, 1, 2, 3, 4$ and $z_i \sim N(0, 1)$ are independent. The coefficients are set as $\beta_i = 3$, $i = 1, 2, \cdots, 15$; $\beta_i = 0$, $i = 16, \cdots, p$.

**Model (iv)** *Factor models*. This model is considered in [17]. Let $\phi_j, j = 1, 2, \cdots, k$ be independent standard normal variables. We set predictors as $x_i = \sum_{j=1}^{k} \phi_j f_{ij} + \eta_i$, where $f_{ij}$ and $\eta_i$ are independent standard normal random variables. The number of factors is chosen as $k = 5$ in the simulation while the coefficients are specified the same as in Model (i).

**Model (v)** $\ell_1$-*ball*. This model takes the same correlation structure as Model (ii), with the coefficients drawn from Dirichlet distribution $\beta \sim Dir\left(\frac{1}{p}, \frac{1}{p}, \cdots, \frac{1}{p}\right) \times 10$. This model is to test the performance under a weakly sparse assumption on $\beta$, since $\beta$ is non-sparse satisfying $\|\beta\|_1 = 10$.

Throughout this section, the performance of all the methods is evaluated in terms of four metrics: the number of false positives (**# FPs**), the number of false negatives (**# FNs**), the mean squared error $\|\hat{\beta} - \beta_*\|_2^2$ (**MSE**) and the computational time (**runtime**). We use `glmnet` [18] to fit *lasso* and choose the tuning parameter using the extended BIC criterion [19] with $\gamma$ fixed at 0.5. For DECO, the features are partitioned randomly in Stage 1 and the tuning parameter $r_1$ is fixed at 1 for **DECO-3**. Since **DECO-2** does not involve any refinement step, we choose $r_1$ to be 10 to aid robustness. The ridge parameter $r_2$ is chosen by 5-fold cross-validation for both **DECO-3** and **lasso-refine**. All

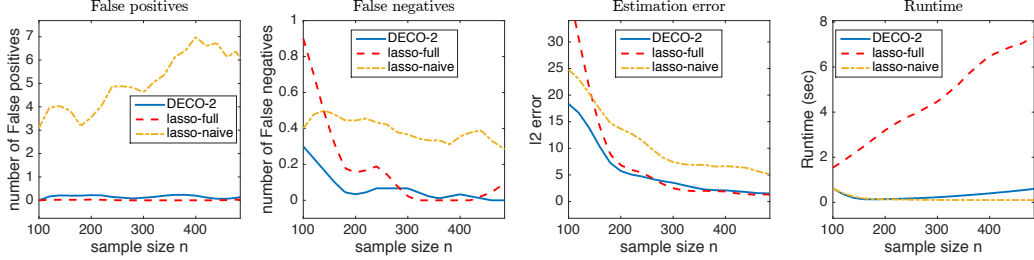

Figure 1: Performance of DECO on one subset with $p = 10,000$ and different $n's$.

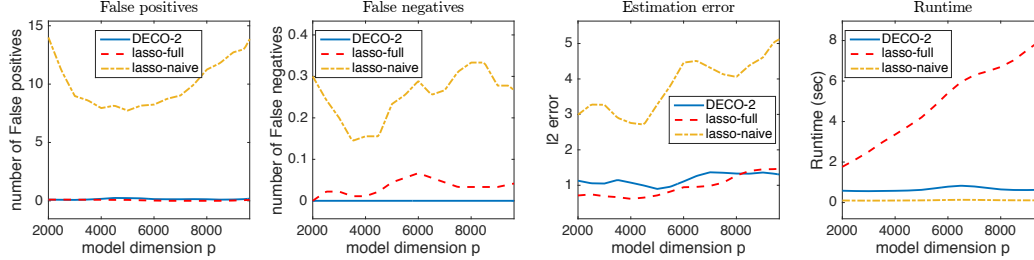

Figure 2: Performance of DECO on one subset with $n = 500$ and different $p's$.

the algorithms are coded and timed in *Matlab* on computers with Intel i7-3770k cores. For any embarrassingly parallel algorithm, we report the preprocessing time plus the longest runtime of a single machine as its runtime.

### 4.1 Monitor DECO on one subset

In this part, using data generated from Model (ii), we illustrate the performance of DECO on one randomly chosen subset after partitioning. The particular subset we examine contains two nonzero coefficients $\beta_1$ and $\beta_2$ with 98 coefficients, randomly chosen, being zero. We either fix $p = 10,000$ and change $n$ from 100 to 500, or fix $n$ at 500 and change $p$ from 2,000 to 10,000 to simulate datasets. We fit **DECO-2**, **lasso-full** and **lasso-naive** to 100 simulated datasets, and monitor their performance on that particular subset. The results are shown in Fig 1 and 2.

It can be seen that, though the sub-model on each subset is mis-specified, DECO performs as if the full dataset were used as its performance is on par with **lasso-full**. On the other hand, **lasso-naive** fails completely. This result clearly highlights the advantage of decorrelation before feature partitioning.

### 4.2 Impact of the subset number $m$

As shown in Theorem 1 and 2, the performance of DECO does not depend on the number of partitions $m$. We verify this property by using Model (ii) again. This time, we fix $p = 10,000$ and $n = 500$, and vary $m$ from 1 to 200. We compare the performance of **DECO-2** and **DECO-3** with **lasso-full** and **lasso-refine**. The averaged results from 100 simulated datasets are plotted in Fig 3. Since $p$ and $n$ are both fixed, **lasso-full** and **lasso-refine** are expected to perform stably over different $m's$. **DECO-2** and **DECO-3** also maintain a stable performance regardless of the value of $m$. In addition, **DECO-3** achieves a similar performance to and sometimes better accuracy than **lasso-refine**, possibly because the irrepresentable condition is satisfied after decorrelation (See the discussions after Theorem 1).

### 4.3 Comprehensive comparison

In this section, we compare all the methods under the five different correlation structures. The model dimension and the sample size are fixed at $p = 10,000$ and $n = 500$ respectively and the number of subsets is fixed as $m = 100$. For each model, we simulate 100 synthetic datasets and record the average performance in Table 1

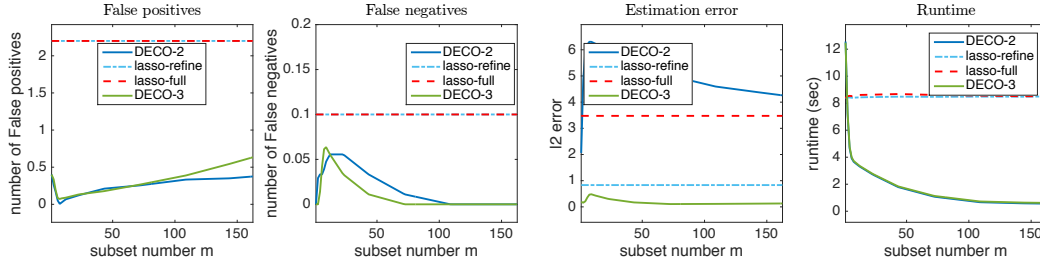

Figure 3: Performance of DECO with different number of subsets.

Table 1: Results for five models with $(n, p) = (500, 10000)$

|       |        | DECO-3 | DECO-2 | lasso-refine | lasso-full | lasso-naive |
|-------|--------|--------|--------|--------------|------------|-------------|
| (i)   | MSE    | 0.102  | 3.502  | 0.104        | 0.924      | 3.667       |
|       | # FPs  | 0.470  | 0.570  | 0.420        | 0.420      | 0.650       |
|       | # FNs  | 0.010  | 0.020  | 0.000        | 0.000      | 0.010       |
|       | Time   | 65.5   | 60.3   | 804.5        | 802.5      | 9.0         |
| (ii)  | MSE    | 0.241  | 4.636  | 1.873        | 3.808      | 171.05      |
|       | # FPs  | 0.460  | 0.550  | 2.39         | 2.39       | 1507.2      |
|       | # FNs  | 0.010  | 0.030  | 0.160        | 0.160      | 1.290       |
|       | Time   | 66.9   | 61.8   | 809.2        | 806.3      | 13.1        |
| (iii) | MSE    | 6.620  | 1220.5 | 57.74        | 105.99     | 1235.2      |
|       | # FPs  | 0.410  | 0.570  | 0.110        | 0.110      | 1.180       |
|       | # FNs  | 0.130  | 0.120  | 3.93         | 3.93       | 0.110       |
|       | Time   | 65.5   | 60.0   | 835.3        | 839.9      | 9.1         |
| (iv)  | MSE    | 0.787  | 5.648  | 11.15        | 6.610      | 569.56      |
|       | # FPs  | 0.460  | 0.410  | 19.90        | 19.90      | 1129.9      |
|       | # FNs  | 0.090  | 0.100  | 0.530        | 0.530      | 1.040       |
|       | Time   | 69.4   | 64.1   | 875.1        | 880.0      | 14.6        |
| (v)   | MSE    | —      | 2.341  | —            | 1.661      | 356.57      |
|       | Time   | —      | 57.5   | —            | 829.5      | 13.3        |

Several conclusions can be drawn from Table 1. First, when all variables are independent as in Model (i), **lasso-naive** performs similarly to **DECO-2** because no decorrelation is needed in this simple case. However, **lasso-naive** fails completely for the other four models when correlations are presented. Second, **DECO-3** achieves the overall best performance. The better estimation error over **lasso-refine** is due to the better variable selection performance, since the irrepresentable condition is not needed for DECO. Finally, **DECO-2** performs similarly to **lasso-full** and the difference is as expected according to the discussions after Theorem 2.

## 5  Real data

We illustrate the competitve performance of DECO via three real datasets that cover a range of high dimensionalities, by comparing **DECO-3** to **lasso-full**, **lasso-refine** and **lasso-naive** in terms of prediction error and computational time. The algorithms are configured in the same way as in Section 4. Although DECO allows arbitrary partitioning over the feature space, for simplicity, we confine our attention to random partitioning. In addition, we perform **DECO-3** multiple times on the same dataset to ameliorate the uncertainty due to the randomness in partitioning.

**Student performance dataset**. We look at one of the two datasets used for evaluating student achievement in two Portuguese schools [20]. The particular dataset used here provides the students' performance in mathematics. The goal of the research is to predict the final grade (range from 0 to 20). The original data set contains 395 students and 32 raw attributes. The raw attributes are recoded as 40 attributes and form 767 features after adding interaction terms. To reduce the conditional number of the feature matrix, we remove features that are constant, giving 741 features. We standardize

all features and randomly partition them into 5 subsets for DECO. To compare the performance of all methods, we use 10-fold cross validation and record the prediction error (mean square error, MSE), model size and runtime. The averaged results are summarized in Table 2. We also report the performance of the null model which predicts the final grade on the test set using the mean final grade in the training set.

**Mammalian eye diseases**. This dataset, taken from [21], was collected to study mammalian eye diseases, with gene expression for the eye tissues of 120 twelve-week-old male F2 rats recorded. One gene coded as TRIM32 responsible for causing Bardet-Biedl syndrome is the response of interest. Following the method in [21], 18,976 probes were selected as they exhibited sufficient signal for reliable analysis and at least 2-fold variation in expressions, and we confine our attention to the top 5,000 genes with the highest sample variance. The 5,000 genes are standardized and partitioned into 100 subsets for DECO. The performance is assessed via 10-fold cross validation following the same approach in Section 5.1. The results are summarized in Table 2. As a reference, we also report these values for the null model.

**Electricity load diagram**. This dataset [22] consists of electricity load from 2011 - 2014 for 370 clients. The data are originally recorded in KW for every 15 minutes, resulting in 14,025 attributes. Our goal is to predict the most recent electricity load by using all previous data points. The variance of the 14,025 features ranges from 0 to $10^7$. To reduce the conditional number of the feature matrix, we remove features whose variances are below the lower $10\%$ quantile (a value of $10^5$) and retain 126,231 features. We then expand the feature sets by including the interactions between the first 1,500 attributes that has the largest correlation with the clients' most recent load. The resulting 1,251,980 features are then partitioned into 1,000 subsets for DECO. Because cross-validation is computationally demanding for such a large dataset, we put the first 200 clients in the training set and the remaining 170 clients in the testing set. We also scale the value of electricity load between 0 and 300, so that patterns are more visible. The results are summarized in Table 2.

Table 2: The results of all methods on the three datasets.

|  | Student Performance | | | Mammalian eye disease | | | Electricity load diagram | | |
|---|---|---|---|---|---|---|---|---|---|
|  | MSE | size | runtime | MSE | size | runtime | MSE | size | runtime |
| **DECO-3** | **3.64** | 1.5 | 37.0 | 0.012 | 4.3 | 9.6 | **0.691** | 4 | 67.9 |
| **lasso-full** | 3.79 | 2.2 | 60.8 | 0.012 | 11 | 139.0 | 2.205 | 6 | 23,515.5 |
| **lasso-refine** | 3.89 | 2.2 | 70.9 | **0.010** | 11 | 139.7 | 1.790 | 6 | 22,260.9 |
| **lasso-naive** | 16.5 | 6.4 | 44.6 | 37.65 | 6.8 | 7.9 | $3.6 \times 10^8$ | 4966 | 52.9 |
| Null | 20.7 | — | — | 0.021 | — | — | 520.6 | — | — |

# 6  Concluding remarks

In this paper, we have proposed an embarrassingly parallel framework named DECO for distributed estimation. DECO is shown to be theoretically attractive, empirically competitive and is straightforward to implement. In particular, we have shown that DECO achieves the same minimax convergence rate as if the full data were used and the rate does not depend on the number of partitions. We demonstrated the empirical performance of DECO via extensive experiments and compare it to various approaches for fitting full data. As illustrated in the experiments, DECO can not only reduce the computational cost substantially, but often outperform the full data approaches in terms of model selection and parameter estimation.

Although DECO is designed to solve large-$p$-small-$n$ problems, it can be extended to deal with large-$p$-large-$n$ problems by adding a sample space partitioning step, for example, using the *message* approach [5]. More precisely, we first partition the large-$p$-large-$n$ dataset in the sample space to obtain $l$ row blocks such that each becomes a large-$p$-small-$n$ dataset. We then partition the feature space of each row block into $m$ subsets. This procedure is equivalent to partitioning the original data matrix $X$ into $l \times m$ small blocks, each with a feasible size that can be stored and fitted in a computer. We then apply the DECO framework to the subsets in the same row block using Algorithm 1. The last step is to apply the *message* method to aggregate the $l$ row block estimators to output the final estimate. This extremely scalable approach will be explored in future work.

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
