[Supplementary Material]

# Supplementary materials to "DECOrrelated feature space partitioning for distributed sparse regression"

Xiangyu Wang
Dept. of Statistical Science
Duke University
wwrechard@gmail.com

David Dunson
Dept. of Statistical Science
Duke University
dunson@stat.duke.edu

Chenlei Leng
Dept. of Statistics
University of Warwick
C.Leng@warwick.ac.uk

Our theory proof consists of two parts. Appendix A provides preliminary results for *lasso*, when strong conditions on the feature matrix are imposed. In Appendix B, we adapt these results to DECO and show that the decorrelated data will automatically satisfy the conditions on the feature matrix even when the original features are highly correlated.

## Appendix A: Review on the lasso theory

Define $Q = \{1, 2, \cdots, p\}$ and let $A^c$ be $Q \setminus A$ for any set $A \subseteq Q$. The following theorem provides deterministic conditions for *lasso* on sup-norm convergence, $\ell_2$-norm convergence and sign consistency.

**Theorem 0.** *Denote the solution to the* lasso *problem as*

$$\hat{\beta} = arg \min_{\beta \in \mathcal{R}^p} \frac{1}{n}\big\|Y - X\beta\big\|_2^2 + 2\lambda_n\|\beta\|_1.$$

*Define $W = Y - X\beta_*$, where $\beta_*$ is the true value of $\beta$. For any arbitrary subset $J \subseteq Q$ (J could be $\emptyset$), if $X$ satisfies that*

1. $M_1 \leq |x_i^T x_i/n| \leq M_2$, *for some $0 < M_1 < M_2$ and all $i$,*

2. $\max_{i \neq j} |x_i^T x_j/n| \leq \min\left\{\frac{1}{\gamma_1 s}, \ \gamma_2\lambda_n^q\right\}$, *for $\gamma_1 > \frac{32}{M_1}, \gamma_2 \geq 0, \ q \geq 0$ and $s = |J|$,*

3. $\|\frac{1}{n}X^T W\|_\infty \leq \lambda_n/2$,

*then any solution to the* lasso *problem satisfies that*

$$\|\hat{\beta} - \beta_*\|_\infty \leq \frac{3M_1\gamma_1 + 51}{2M_1(M_1\gamma_1 - 7)}\lambda_n + \frac{4M_1\gamma_1\gamma_2 + 36\gamma_2}{M_1(M_1\gamma_1 - 7)}\|\beta_{*J^c}\|_1\lambda_n^q + \frac{8\sqrt{3\gamma_2}}{M_1\sqrt{M_1\gamma_1 - 7}}\|\beta_{*J^c}\|_1^{\frac{1}{2}}\lambda_n^{\frac{1+q}{2}},$$

*where $\beta_{*J^c}$ is the sub-vector of $\beta_*$ consisting of coordinates in $J^c$ and*

$$\|\hat{\beta} - \beta_*\|_2^2 \leq \frac{18\gamma_1^2 s\lambda_n^2}{(M_1\gamma_1 - 32)^2} + 6\lambda_n\|\beta_{*J^c}\|_1 + 32\gamma_2\lambda_n^q\|\beta_{*J^c}\|_1^2.$$

*Furthermore, if $\beta_{*J^c} = 0$ and $\min_{k \in J}|\beta_{*k}| \geq \frac{2}{M_1}\lambda_n$, then the solution is unique and sign consistent, that is,*

$$sign(\hat{\beta}_k) = sign(\beta_{*k}), \ \forall k \in J \quad and \quad \hat{\beta}_k = 0, \ \forall k \in J^c.$$

Theorem 0 partly extends the results in Bickel et al. (2009) and Lounici (2008). The proof is provided in Appendix A. Theorem 0 can lead to some useful results. In particular, we investigate two types of models when $\beta_*$ is either exactly sparse or in an $l_r$-ball defined as $\mathbb{B}(r, R) = \{v \in \mathcal{R}^p : \sum_{k=1}^p |v_k|^r \leq R\}$. For the exactly sparse model, we have the following result.

**Corollary 1** (s-sparse). *Assume that $\beta_* \in \mathcal{R}^p$ is an s-sparse vector with $J$ containing all non-zero indices. If Condition 1 and 3 in Theorem 0 hold and $\max_{i \neq j} |x_i^T x_j/n| \leq \frac{1}{\gamma_1 s}$ for some $\gamma_1 > 32/M_1$, then we have*

$$\|\hat{\beta} - \beta_*\|_\infty \leq \frac{3M_1\gamma_1 + 51}{2M_1(M_1\gamma_1 - 7)}\lambda_n \quad and \quad \|\hat{\beta} - \beta_*\|_2^2 \leq \frac{18\gamma_1^2 s\lambda_n^2}{(M_1\gamma_1 - 32)^2}.$$

*Further, if $\min_{k \in J} |\beta_k| \geq \frac{2}{M_1}\lambda_n$, then $\hat{\beta}$ is sign consistent.*

The sup-norm convergence in Corollary 1 resembles the results in Lounici (2008). For the $l_r$-ball we have

**Corollary 2** ($l_r-$ ball). *Assume $\beta_* \in \mathbb{B}(r, R)$. If condition 1 and 3 in Theorem 0 hold and $\max_{i \neq j} |x_i^T x_j/n| \leq \frac{\lambda_n^r}{\gamma_1 R}$ for some $\gamma_1 > 32/M_1$, then we have*

$$\|\hat{\beta} - \beta_*\|_\infty \leq \left(\frac{3M_1\gamma_1 + 51}{2M_1(M_1\gamma_1 - 7)} + \frac{4M_1\gamma_1 + 36}{M_1(M_1\gamma_1 - 7)}\right)\lambda_n + \frac{8\sqrt{3}}{M_1\sqrt{M_1\gamma_1 - 7}}\lambda_n,$$

$$\|\hat{\beta} - \beta_*\|_2^2 \leq \left(\frac{18\gamma_1^2}{(M_1\gamma_1 - 32)^2} + 38\right)R\lambda_n^{2-r}.$$

## A.1 Proof of the $\ell_2$ and $\ell_\infty$ convergence

We first need the following lemmas

**Lemma 0.** *Assuming the Condition 3 in Theorem 0, and defining $\Delta = \hat{\beta} - \beta_*$, where $\hat{\beta}$ is the solution to the lasso problem and $\beta_*$ is the true value, then for any set $J \subseteq Q$ (J could be $\emptyset$), where $Q = \{1, 2, \cdots, p\}$, we have*

$$\|\Delta_{J^c}\|_1 \leq 3\|\Delta_J\|_1 + 4\|\beta_{*J^c}\|_1, \tag{1}$$

*where $\Delta_J$ denotes a sub-vector of $\Delta$ containing coordinates whose indexes belong to $J$ and $\|\Delta_\emptyset\|_1 = 0$.*

*Proof of Lemma 0.* We follow the proof in Bickel et al. (2009) and Lounici (2008). Define $\hat{S}(\hat{\beta}) = \{k : \hat{\beta}_k \neq 0\}$. The sufficient and necessary condition (also known as the KKT conditions) for $\hat{\beta}$ to minimize the *lasso* problem is that

$$\frac{1}{n}(x_i^T Y - x_i^T X\hat{\beta}) = \lambda_n sign(\hat{\beta}_i), \text{ for } i \in \hat{S}(\hat{\beta})$$

$$\frac{1}{n}|x_i^T Y - x_i^T X\hat{\beta}| \leq \lambda_n, \text{ for } i \notin \hat{S}(\hat{\beta}).$$

Therefore, regardless of $\hat{S}(\hat{\beta})$, the minimizer $\hat{\beta}$ always satisfies that

$$\frac{1}{n}\|X^T Y - X^T X\hat{\beta}\|_\infty \leq \lambda_n.$$

Noticing that $Y = X\beta_* + W$ and $\frac{1}{n}\|X^T W\|_\infty \leq \lambda_n/2$, we have

$$\frac{1}{n}\|X^T X(\beta_* - \hat{\beta})\|_\infty \leq \frac{3}{2}\lambda_n. \tag{2}$$

At the same time, using the optimality of *lasso* we have

$$\frac{1}{n}\|Y - X\hat{\beta}\|_2^2 + 2\lambda_n\|\hat{\beta}\|_1 \leq \frac{1}{n}\|Y - X\beta_*\|_2^2 + 2\lambda_n\|\beta_*\|_1 = \frac{1}{n}\|W\|_2^2 + 2\lambda_n\|\beta_*\|_1,$$

which implies

$$2\lambda_n\|\hat{\beta}\|_1 \leq 2\lambda_n\|\beta_*\|_1 + \frac{1}{n}\|W\|_2^2 - \frac{1}{n}\|Y - X\hat{\beta}\|_2^2$$
$$= 2\lambda_n\|\beta_*\|_1 + \frac{1}{n}\|W\|_2^2 - \frac{1}{n}\|X\beta_* - X\hat{\beta} + W\|_2^2$$
$$\leq 2\lambda_n\|\beta_*\|_1 + \left|2(\hat{\beta} - \beta_*)\frac{X^T W}{n}\right|.$$

Using $\|\frac{1}{n}X^T W\|_\infty \leq \lambda_n/2$, we know that

$$2\lambda_n\|\hat{\beta}\|_1 \leq 2\lambda_n\|\beta_*\|_1 + \lambda_n\|\hat{\beta} - \beta_*\|_1,$$

i.e., we have

$$2\|\hat{\beta}\|_1 \leq 2\|\beta_*\|_1 + \|\hat{\beta} - \beta_*\|_1 = 2\|\beta_*\|_1 + \|\Delta\|_1, \tag{3}$$

Let $J$ be any arbitrary subset of $Q$, we have

$$2\|\Delta_{J^c}\|_1 = 2\|\hat{\beta}_{J^c} - \beta_{*J^c}\|_1 \leq 2\|\hat{\beta}_{J^c}\|_1 + 2\|\beta_{*J^c}\|_1. \tag{4}$$

Now if $J = \emptyset$, using (3) and (4) we have

$$2\|\Delta\|_1 = 2\|\Delta_{J^c}\|_1 \leq 2\|\hat{\beta}_{J^c}\|_1 + 2\|\beta_{*J^c}\|_1 = 2\|\hat{\beta}\|_1 + 2\|\beta_*\|_1 \leq 4\|\beta_*\|_1 + \|\Delta\|_1.$$

This gives that

$$\|\Delta_{J^c}\|_1 = \|\Delta\|_1 \leq 4\|\beta_*\|_1 = 3\|\Delta_J\|_1 + 4\|\beta_{*J^c}\|_1.$$

For $J \neq \emptyset$, because $\ell_1$ norm is decomposable, i.e., $\|\hat{\beta}\|_1 = \|\hat{\beta}_J\|_1 + \|\hat{\beta}_{J^c}\|_1$, using (3), we have

$$2\|\hat{\beta}_{J^c}\|_1 + 2\|\beta_{*J^c}\|_1 = 2\|\hat{\beta}\|_1 - 2\|\hat{\beta}_J\|_1 + 2\|\beta_{*J^c}\|_1$$
$$\leq 2\|\beta_*\|_1 + \|\Delta\|_1 - 2\|\hat{\beta}_J\|_1 + 2\|\beta_{*J^c}\|_1$$
$$= 2\|\beta_{*J}\|_1 + 2\|\beta_{*J^c}\|_1 + \|\Delta_J\|_1 + \|\Delta_{J^c}\|_1 - 2\|\hat{\beta}_J\|_1 + 2\|\beta_{*J^c}\|_1$$
$$= 2(\|\beta_{*J}\|_1 - \|\hat{\beta}_J\|_1) + \|\Delta_J\|_1 + \|\Delta_{J^c}\|_1 + 4\|\beta_{*J^c}\|_1$$
$$\leq 3\|\Delta_J\|_1 + \|\Delta_{J^c}\|_1 + 4\|\beta_{*J^c}\|_1,$$

where the second inequality is due to (3). Thus, combining the above result with (4) we have proved that

$$\|\Delta_{J^c}\|_1 \leq 3\|\Delta_J\|_1 + 4\|\beta_{*J^c}\|_1.$$

$\square$

**Lemma 1.** *Assume the Condition 1 and 2 in Theorem 0. For any $J \subseteq \{1, 2, \cdots, p\}$ (J could be $\emptyset$) and $|J| \leq s$ and any $v \in \mathcal{R}^p$ such that $\|v_{J^c}\|_1 \leq c_0\|v_J\|_1 + c_1\|\beta_{*J^c}\|_1$, we have*

$$\frac{1}{n}\|Xv\|_2^2 \geq \left(M_1 - \frac{1 + 2c_0}{\gamma_1}\right)\|v_J\|_2^2 - 2c_1\gamma_2\sqrt{s}\lambda_n^q\|\beta_{*J^c}\|_1\|v_J\|_2, \tag{5}$$

*where $v_J$ denotes a sub-vector of $v$ containing coordinates whose indexes belong to $J$.*

*Proof of Lemma 1.* When $J = \emptyset$, the result is straightforward and thus omitted. Assume $|J| > 0$. For convenience, we define $\tilde{v}$ to be the vector that extends $v_J$ to $p$-dimensional by adding zero coodinates, i.e.,

$$\tilde{v}_i = v_i \quad \text{if } i \in J$$
$$\tilde{v}_i = 0 \quad \text{if } i \notin J$$

We use $v_J^{(i)}$ to denote the $i^{th}$ coordinate of $v_J$. For any $J \subseteq \{1, 2, \cdots, p\}$ with $|J| = s$ and any $v \in \mathcal{R}^p$ such that $\|v_{J^c}\|_1 \le c_0 \|v_J\|_1 + c_1 \|\beta_{*J^c}\|_1$, we have

$$\frac{\|X\tilde{v}\|_2^2}{n\|v_J\|_2^2} = \frac{\tilde{v}^T(X^TX/n - M_1 I_p)\tilde{v}}{\|v_J\|_2^2} + M_1$$

$$= M_1 + \frac{\sum_{i=1}^p (x_i^T x_i/n - M_1)|\tilde{v}_i|^2 + \sum_{i \ne j}(x_i^T x_j/n)\tilde{v}_i \tilde{v}_j}{\|v_J\|_2^2}$$

$$= M_1 + \frac{\sum_{i \in J}(x_i^T x_i/n - M_1)|v_J^{(i)}|^2 + \sum_{i \ne j \in J}(x_i^T x_j/n)v_J^{(i)} v_J^{(i)}}{\|v_J\|_2^2}$$

$$\ge M_1 - \frac{1}{\gamma_1 s} \sum_{i \ne j} \frac{v_J^{(i)} v_J^{(j)}}{\|v_J\|_2^2} \ge M_1 - \frac{1}{\gamma_1 s} \frac{\|v_J\|_1^2}{\|v_J\|_2^2}.$$

Notice that $\|v_J\|_1^2 \le s\|v_J\|_2^2$ because $|J| \le s$. Thus, we have

$$\frac{1}{n}\|Xv\|_2^2 \ge \frac{1}{n}\|X\tilde{v}\|_2^2 + 2\tilde{v}^T(\frac{1}{n}X^TX)(v - \tilde{v})$$

$$\ge M_1\|v_J\|_2^2 - \frac{1}{\gamma_1 s}\|v_J\|_1^2 - 2\max_{i \ne j}\frac{1}{n}|x_i^T x_j|\|v_J\|_1\|v_{J^c}\|_1$$

$$\ge \left(M_1 - \frac{1}{\gamma_1}\right)\|v_J\|_2^2 - 2c_0\max_{i \ne j}\frac{1}{n}|x_i^T x_j|\|v_J\|_1^2 - 2c_1\max_{i \ne j}\frac{1}{n}|x_i^T x_j|\|\beta_{*J^c}\|_1\|v_J\|_1$$

$$\ge \left(M_1 - \frac{1}{\gamma_1}\right)\|v_J\|_2^2 - \frac{2c_0}{\gamma_1 s}\|v_J\|_1^2 - 2c_1\gamma_2\lambda_n^q\beta_{*J^c}\|_1\|v_J\|_1$$

$$\ge \left(M_1 - \frac{1 + 2c_0}{\gamma_1}\right)\|v_J\|_2^2 - 2c_1\gamma_2\sqrt{s}\lambda_n^q\|\beta_{*J^c}\|_1\|v_J\|_2.$$

$\square$

**Lemma 2.** *Assume the Condition 1 and 2 in Theorem 0. For any $J \subseteq \{1, 2, \cdots, p\}$ (J could be $\emptyset$) and $|J| \le s$ and any $v \in \mathcal{R}^p$ such that $\|v_{J^c}\|_1 \le c_0\|v_J\|_1 + c_1\|\beta_{*J^c}\|_1$, we have*

$$\frac{1}{n}\|Xv\|_2^2 \ge \left(M_1 - \frac{2(1 + c_0)^2}{\gamma_1}\right)\|v\|_2^2 - 2c_1^2\lambda_n^q\|\beta_{*J^c}\|_1^2. \tag{6}$$

*where $v_J$ denotes a sub-vector of $v$ containing coordinates whose indexes belong to $J$.*

*Proof of Lemma 2.* Different from Lemma 1, we have

$$\frac{1}{n}\|Xv\|_2^2 \geq \sum_{i \in Q} \frac{1}{n}\|x_i\|_2^2 v_i^2 + \sum_{i \neq j \in Q} \frac{1}{n} x_i^T x_j v_i v_j$$

$$\geq M_1\|v\|_2^2 - \max_{i \neq j} \frac{1}{n}|x_i^T x_j|\|v\|_1^2 = M_1\|v\|_2^2 - \max_{i \neq j} \frac{1}{n}|x_i^T x_j|(\|v_J\|_1 + \|v_{J^c}\|_1)^2$$

$$\geq M_1\|v\|_2^2 - \max_{i \neq j} \frac{1}{n}|x_i^T x_j|\left((1 + c_0)\|v_J\|_1 + c_1\|\beta_{*J^c}\|_1\right)^2$$

$$\geq M_1\|v\|_2^2 - 2\max_{i \neq j} \frac{1}{n}|x_i^T x_j|(1 + c_0)^2\|v_J\|_1^2 - 2\max_{i \neq j} \frac{1}{n}|x_i^T x_j|c_1^2\|\beta_{*J^c}\|_1^2$$

$$\geq M_1\|v\|_2^2 - \frac{2(1 + c_0)^2}{\gamma_1}\|v_J\|_2^2 - 2c_1^2\gamma_2\lambda_n^q\|\beta_{*J^c}\|_1^2$$

$$\geq \left(M_1 - \frac{2(1 + c_0)^2}{\gamma_1}\right)\|v\|_2^2 - 2c_1^2\gamma_2\lambda_n^q\|\beta_{*J^c}\|_1^2$$

$\square$

Now, We turn to the proof of $\ell_2$ and $\ell_\infty$ convergence in Theorem 0.

**(Partial) proof of Theorem 0.** According to Lemma 0, 1, 2 and (1), (2) and (5), we have

$$\left\|\frac{1}{n}X^T X\Delta\right\|_\infty \leq \frac{3}{2}\lambda_n \tag{7}$$

and

$$\|\Delta\|_1 \leq 4\|\Delta_J\|_1 + 4\|\beta_{*J^c}\|_1 \leq 4\sqrt{s}\|\Delta_J\|_2 + 4\|\beta_{*J^c}\|_1 \tag{8}$$

and

$$\frac{1}{n}\|X\Delta\|_2^2 \geq \left(M_1 - \frac{7}{\gamma_1}\right)\|\Delta_J\|_2^2 - 8\gamma_2\sqrt{s}\lambda_n^q\|\beta_{*J^c}\|_1\|\Delta_J\|_2. \tag{9}$$

Using Equations (7) and (8), we have

$$\frac{1}{n}\|X\Delta\|_2^2 \leq \|\frac{1}{n}X^T X\Delta\|_\infty\|\Delta\|_1 \leq 6\lambda_n\sqrt{s}\|\Delta_J\|_2 + 6\lambda_n\|\beta_{*J^c}\|_1,$$

which combining with (9) implies that

$$\left(M_1 - \frac{7}{\gamma_1}\right)\|\Delta_J\|_2^2 - 2(3\sqrt{s}\lambda_n + 4\gamma_2\sqrt{s}\lambda_n^q\|\beta_{*J^c}\|_1)\|\Delta_J\|_2 - 6\lambda_n\|\beta_{*J^c}\|_1 \leq 0$$

This is a quadratic form and with some simple algebra, we get a loose solution to the quadratic inequality

$$\frac{1}{2}\left(M_1 - \frac{7}{\gamma_1}\right)\|\Delta_J\|_2^2 \leq \frac{2(3\sqrt{s}\lambda_n + 4\gamma_2\sqrt{s}\lambda_n^q\|\beta_{*J^c}\|_1)^2}{M_1 - \frac{7}{\gamma_1}} + 6\lambda_n\|\beta_{*J^c}\|_1,$$

thus

$$\|\Delta_J\|_2^2 \leq \frac{72\gamma_1^2 s}{(M_1\gamma_1 - 7)^2}\lambda_n^2 + \frac{192\gamma_1^2\gamma_2^2\|\beta_{*J^c}\|_1^2 s}{(M_1\gamma_1 - 7)^2}\lambda_n^{2q} + \frac{12\gamma_1\|\beta_{*J^c}\|_1}{M_1\gamma_1 - 7}\lambda_n,$$

and thus

$$\|\Delta_J\|_2 \leq \sqrt{\frac{72\gamma_1^2 s}{(M_1\gamma_1 - 7)^2}\lambda_n^2 + \frac{192\gamma_1^2\gamma_2^2\|\beta_{*J^c}\|_1^2 s}{(M_1\gamma_1 - 7)^2}\lambda_n^{2q} + \frac{12\gamma_1\|\beta_{*J^c}\|_1}{M_1\gamma_1 - 7}\lambda_n}$$

$$\leq \frac{6\sqrt{2}\gamma_1}{M_1\gamma_1 - 7}\sqrt{s}\lambda_n + \frac{8\sqrt{3}\gamma_1\gamma_2\|\beta_{*J^c}\|_1}{M_1\gamma_1 - 7}\sqrt{s}\lambda_n^q + \frac{2\sqrt{3}\gamma_1^{\frac{1}{2}}\|\beta_{*J^c}\|_1^{\frac{1}{2}}}{\sqrt{M_1\gamma_1 - 7}}\lambda_n^{\frac{1}{2}} \quad (10)$$

Similarly, for $\|\Delta\|_2^2$, using (6) we have

$$\left(M_1 - \frac{32}{\gamma_1}\right)\|\Delta\|_2^2 - 32\gamma_2\lambda_n^q\|\beta_{*J^c}\|_1^2 \leq \frac{1}{n}\|X\Delta\|_2^2 \leq 6\lambda_n\sqrt{s}\|\Delta_J\|_2 + 6\lambda_n\|\beta_{*J^c}\|_1.$$

Noticing that $\|\Delta_J\|_2 \leq \|\Delta\|_2$, we can solve the quadratic inequality and obtain that

$$\|\Delta\|_2^2 \leq \frac{18\gamma_1^2 s\lambda_n^2}{(M_1\gamma_1 - 32)^2} + 6\lambda_n\|\beta_{*J^c}\|_1 + 32\gamma_2\lambda_n^q\|\beta_{*J^c}\|_1^2. \quad (11)$$

For the sup-norm, we make use of (10). Notice that

$$e_j^T\frac{X^TX}{n}\Delta = \frac{x_j^TX}{n}\Delta = \frac{\|x_j\|_2^2}{n}\Delta_j + \sum_{i\neq j}\frac{x_i^Tx_j}{n}\Delta_i$$

which combning with (7) and (8) implies that

$$\frac{\|x_j\|_2^2}{n}|\Delta_j| \leq \left|e_j^T\frac{X^TX}{n}\Delta\right| + \left|\sum_{i\neq j}\frac{x_i^Tx_j}{n}\Delta_i\right| \leq \|\frac{1}{n}X^TX\Delta\|_\infty + \max_{i\neq k}\frac{1}{n}|x_i^Tx_k|\|\Delta\|_1$$

$$\leq \frac{3}{2}\lambda_n + 4\max_{i\neq k}\frac{1}{n}|x_i^Tx_k|\sqrt{s}\|\Delta_J\|_2 + 4\max_{i\neq k}\frac{1}{n}|x_i^Tx_k|\|\beta_{*J^c}\|_1$$

Note that $\max_{i\neq k}\frac{1}{n}|x_i^Tx_k| \leq \min\{\frac{1}{\gamma_1 s},\ \gamma_2\lambda_n^q\}$ also implies that $\max_{i\neq k}\frac{1}{n}|x_i^Tx_k| \leq \sqrt{\frac{\gamma_2\lambda_n^q}{\gamma_1 s}}$. Therefore, using result in (10) we have

$$M_1\|\Delta\|_\infty \leq \frac{3}{2}\lambda_n + \frac{24\sqrt{2}}{M_1\gamma_1 - 7}\lambda_n + \frac{32\sqrt{3}\gamma_2}{M_1\gamma_1 - 7}\|\beta_{*J^c}\|_1\lambda_n^q + \frac{8\sqrt{3}\gamma_2^{\frac{1}{2}}}{\sqrt{M_1\gamma_1 - 7}}\|\beta_{*J^c}\|_1^{\frac{1}{2}}\lambda_n^{\frac{1+q}{2}} + 4\gamma_2\|\beta_{*J^c}\|_1\lambda_n^q,$$

which yields,

$$\|\Delta\|_\infty \leq \frac{3M_1\gamma_1 + 51}{2M_1(M_1\gamma_1 - 7)}\lambda_n + \frac{4M_1\gamma_1\gamma_2 + 36\gamma_2}{M_1(M_1\gamma_1 - 7)}\|\beta_{*J^c}\|_1\lambda_n^q + \frac{8\sqrt{3\gamma_2}}{M_1\sqrt{M_1\gamma_1 - 7}}\|\beta_{*J^c}\|_1^{\frac{1}{2}}\lambda_n^{\frac{1+q}{2}}.$$

This completes the proof. □

## A.2 Proof of the sign consistency

Our conclusion on sign consistency is stated as follows

**Theorem 1.** *Let $J$ be the set containing indexes of all the nonzero coefficients. Assume all the conditions in Theorem 0. In addition, if the following conditions hold*

$$\min_{k\in J}|\beta_k| \geq \frac{2}{M_1}\lambda_n,$$

*then the solution to the* lasso *is unique and satisfies the sign consistency, i.e,*

$$sign(\hat{\beta}_k) = sign(\beta_{*k}), \ \forall k \in J \quad and \quad \hat{\beta}_k = 0, \ \forall k \in J^c.$$

Here we use the primal-dual witness (PDW) approach (Wainwright, 2009) to prove sign consistency. The PDW approach works on the following two terms

$$Z_k = \frac{1}{n\lambda_n} x_k^T \Pi_{X_J^\perp} W + \frac{1}{n} x_k^T X_J \left( \frac{1}{n} X_J^T X_J \right)^{-1} \check{z}_J,$$

where $\Pi_A$ is the projection on to the linear space spanned by the vectors in $A$ and

$$\Delta_k = e_k^T \left( \frac{1}{n} X_J^T X_J \right)^{-1} \left( \frac{1}{n} X_J^T W - \lambda_n sign(\beta_{*J}) \right),$$

for which Wainwright (2009) proves the following lemma

**Lemma 3.** *(Wainwright, 2009) If $Z_k$ and $\Delta_k$ satisfy that*

$$sign(\beta_{*k} + \Delta_k) = sign(\beta_{*k}), \ \forall k \in J \quad and \quad |Z_k| < 1, \ \forall k \in J^c,$$

*then the optimal solution to* lasso *is unique and satisfies the sign consistency, i.e.,*

$$sign(\hat{\beta}_k) = sign(\beta_{*k}), \ \forall k \in J \quad and \quad \hat{\beta}_k = 0, \ \forall k \in J^c.$$

Therefore, we just need to verify the two conditions in Lemma 3 for Theorem 1. Before we proceed to prove Theorem 1, we state another lemma that is needed for the proof.

**Lemma 4.** *(Varah, 1975) Let $A$ be a strictly diagonally dominant matrix and define $\delta = \min_k(|A_{kk}| - \sum_{j \neq k} |A_{kj}|) > 0$, then we have*

$$\|A^{-1}\|_\infty \leq \delta^{-1},$$

*where $\|A\|_\infty$ is the maximum of the row sums of $A$.*

**Proof of Theorem 1.** We first bound $|Z_k|$ for $k \in J^c$. Notice the first term in $Z_k$ follows that

$$\frac{1}{n\lambda_n} x_k^T \Pi_{X_J^\perp} W = \frac{1}{n\lambda_n} x_k^T W - \frac{1}{n\lambda_n} x_k^T X_J (X_J^T X_J)^{-1} X_J^T W,$$

where $\frac{1}{n\lambda_n} x_k^T W$ follows

$$\left| \frac{1}{n\lambda_n} x_k^T W \right| \leq \frac{1}{\lambda_n} \| \frac{1}{n} X^T W \|_\infty \leq \frac{1}{2}$$

and $\frac{1}{n\lambda_n} x_k^T X_J (X_J^T X_J)^{-1} X_J^T W$ follows

$$\left| \frac{1}{n\lambda_n} x_k^T X_J (X_J^T X_J)^{-1} X_J^T W \right| \leq \frac{1}{\lambda_n} \| \frac{1}{n} x_k^T X_J \|_1 \| (X_J^T X_J)^{-1} X_J^T W \|_\infty$$

From Condition 2 in Theorem 0, we know that

$$\| \frac{1}{n} x_k^T X_J \|_1 \leq \sum_{j \in J} \frac{1}{n} |x_k^T x_j| \leq \frac{1}{\gamma_1}$$

and using Lemma 4 we have

$$\|(X^T X/n)^{-1}\|_\infty = \max_{k \in Q} \|e_k^T (X_J^T X_J/n)^{-1}\|_1 \le (M_1 - 1/\gamma_1)^{-1}.$$

Thus, we have

$$\frac{1}{\lambda_n}\|(X_J^T X_J)^{-1} X_J^T W\|_\infty \le \frac{1}{\lambda_n}\|(X_J^T X_J/n)^{-1}\|_\infty \frac{1}{n} X_J^T W\|_\infty \le \frac{\gamma_1}{2(M_1 \gamma_1 - 1)}.$$

Together, the first term can be bounded as

$$\left| \frac{1}{n\lambda_n} x_k^T \Pi_{X_J^\perp} W \right| \le \frac{1}{2} + \frac{1}{2(M_1 \gamma_1 - 1)}. \tag{12}$$

The second term can be bounded similarly as the first term, i.e.,

$$\left| \frac{1}{n} x_k^T X_J (X_J^T X_J)^{-1} \check{z}_J \right| \le \|\frac{1}{n} x_k^T X_J\|_1 \|(X_J^T X_J)^{-1} \check{z}_J\|_\infty \le \frac{1}{M_1 \gamma_1 - 1},$$

Therefore, we have

$$|Z_k| \le \frac{1}{2} + \frac{3}{2(M_1 \gamma_1 - 1)}.$$

It is easy to see that when $\gamma_1 > 32/M_1$, we have

$$|Z_k| < 1, \ \forall k \in J^c$$

and completes the proof for $Z_k$. We now turn our attention to $\Delta_k$ and check whether $sign(\beta_{*k}) = sign(\beta_{*k} + \Delta_k)$. For $\Delta_k$, we have

$$
\begin{aligned}
|\Delta_k| &= \left| e_k^T \left( \frac{1}{n} X_J^T X_J \right)^{-1} \left( \frac{1}{n} X_J^T W - \lambda_n sign(\beta_{*J}) \right) \right| \\
&\le \left| e_k^T \left( \frac{1}{n} X_J^T X_J \right)^{-1} \frac{X_J^T W}{n} \right| + \lambda_n \left\| \left( \frac{1}{n} X_J^T X_J \right)^{-1} \right\|_\infty \\
&\le \left\| \left( \frac{1}{n} X_J^T X_J \right)^{-1} \right\|_\infty \|X_J^T W/n\|_\infty + \lambda_n \left\| \left( \frac{1}{n} X_J^T X_J \right)^{-1} \right\|_\infty \\
&\le \frac{\gamma_1}{2(M_1 \gamma_1 - 1)} \lambda_n + \frac{\gamma_1}{M_1 \gamma_1 - 1} \lambda_n \\
&= \frac{3\gamma_1}{2(M_1 \gamma_1 - 1)} \lambda_n.
\end{aligned}
$$

Thus, with the conditions in Theorem 2, we have

$$|\Delta_k| \le \frac{3\gamma_1}{2(M_1 \gamma_1 - 1)} \lambda_n = \frac{3}{2(M_1 - 1/\gamma_1)} \lambda_n < \frac{2}{M_1} \lambda_n.$$

To meet the requirement $sign(\beta_{*k}) = sign(\beta_{*k} + \Delta_k)$, we just need $\min_{k \in J} |\beta_k| \ge \frac{2}{M_1} \lambda_n$ and this completes the proof. □

## A.3: Proof of Corollary 1 and 2

To prove the two corollaries, we just need to adapt the magnitude of $\max_{i \ne j} \frac{1}{n}|x_i^T x_j|$ to the correct order.

**Proof of Corollary 1 and 2.** To prove Corollary 1, we just need to take $\gamma_2$ arbitrarily large and $q = 1$. The result follows immediately from Theorem 0.

To prove Corollary 2, we first determine the set $J$ by taking the larger signals as follows

$$J = \{k : |\beta_k| \geq \lambda_n\}.$$

Then the size of $J$ can be bounded as

$$s = |J| \leq \frac{R}{\lambda_n^r}$$

and the size of $\|\beta_* J^c\|_1$ can be bounded as

$$\|\beta_* J^c\|_1 = \sum_{k \in J^c} |\beta_* k| \leq \lambda_n^{1-r} \sum_{k \in J^c} |\beta_* k|^r \leq R\lambda_n^{1-r}.$$

Now we take $\gamma_2 = 1/\|\beta_* J^c\|_1$ and $q = 1$, then the bound on $\max_{i \neq j} \frac{1}{n}|x_i^T x_j|$ becomes

$$\max_{i \neq j} \frac{1}{n}|x_i^T x_j| \leq \min\left\{\frac{1}{\gamma_1 s}, \frac{\lambda_n}{\|\beta_* J^c\|_1}\right\} \leq \min\left\{\frac{\lambda_n^r}{\gamma_1 R}, \frac{\lambda_n^r}{R}\right\} \leq \frac{\lambda_n^r}{\gamma_1 R},$$

which completes the proof. $\qquad\square$

# Appendix B: Proof Theorem 1 and 2

To prove the two theorems, we just need to verify the three conditions for DECO. To verify Condition 1 and Condition 2 in Theorem 0, we cite a result from Wang et al. (2015) which proves the boundedness of $M_1$ and $M_2$ and that $\max_{i \neq j} |\tilde{x}_i^T \tilde{x}_j^T|/n$ is small.

**Lemma 5.** *Assuming $X \sim N(0, \Sigma)$ and $p > c_0 n$ for some $c_0 > 1$, we have that for any $C > 0$, there exists some constant $0 < c_1 < 1 < c_2$ and $c_3 > 0$ such that for any $i \neq j \in Q$*

$$\mathbb{P}\left(\frac{1}{n}|\tilde{x}_i|_2^2 < \frac{c_1 c_*}{c^*}\right) \leq 2e^{-Cn}, \quad \mathbb{P}\left(\frac{1}{n}|\tilde{x}_i|_2^2 > \frac{c_2 c^*}{c_*}\right) \leq 2e^{-Cn},$$

*and*

$$\mathbb{P}\left(\frac{1}{n}|\tilde{x}_i^T \tilde{x}_j| > \frac{c_4 c^* t}{c_*} \frac{1}{\sqrt{n}}\right) \leq 5e^{-Cn} + 2e^{-t^2/2},$$

*for any $t > 0$, where $c_4 = \sqrt{\frac{c_2(c_0 - c_1)}{c_3(c_0 - 1)}}$ and $c_*, c^*$ are the smallest and largest eigenvalues of $\Sigma$.*

Verifying Condition 3 is the key to the whole proof. Different from the conventional setting, $\tilde{W}$ now contains non-zero signals that are not independent from the predictors. This requires us to accurately capture the behavior of the following two terms $\max_{k \in Q} \left|\frac{1}{n}\tilde{x}_k^T \tilde{X}^{(-k)}\beta_*^{(-k)}\right|$ and $\max_{k \in Q} \left|\frac{1}{n}\tilde{x}_k^T \tilde{\varepsilon}\right|$, for which we have

**Lemma 6.** *Assume that $\varepsilon$ is a sub-Gaussian variable with a $\psi_2$ norm of $\sigma$ and $X \sim N(0, \Sigma)$. Define $\sigma_0^2 = var(Y)$. If $p > c_0 n$ for some $c_0 > 1$, then we have for any $t > 0$*

$$\mathbb{P}\left(\max_{k \in Q} \frac{1}{n}|\tilde{x}_k^T \tilde{\varepsilon}| > \frac{\sigma t}{\sqrt{n}}\right) \leq 2p\exp\left(-\frac{c_* c_0^2}{2c^* c_2(1 - c_0)^2}t^2\right) + 4pe^{-Cn},$$

$$\mathbb{P}\left(\max_{k \in Q} \frac{1}{n}|\tilde{x}_k^T \tilde{X}^{(-k)}\beta_*^{(-k)}| \geq \frac{\sqrt{\sigma_0^2 - \sigma^2}t}{\sqrt{n}}\right) \leq 2p\exp\left(-\frac{c_*^3}{2c_4^2 c^{*2}}t^2\right) + 5pe^{-Cn},$$

where $C, c_1, c_2, c_4, c_*, c^*$ are defined in Lemma 5.

## B.1: Proof of Lemma 5 and 6

Lemma 5 and the first part of 6 are existing results from Wang et al. (2015) and Wang and Leng (2015). We focus on proving the second part of Lemma 6.

**Proof of Lemma 5 and 6.** Lemma 5 follows immediately from Lemma 3 in Wang et al. (2015) and the first part of Lemma 6 follows Lemma 4 in Wang et al. (2015).

To prove the second part of Lemma 6, we first define $H = X^T (XX^T)^{-\frac{1}{2}}$. When $X \sim N(0, \Sigma)$, $H$ follows the $MACG(\Sigma)$ distribution as indicated in Lemma 3 in Wang et al. (2015) and Theorem 1 in Wang and Leng (2015). For simplicity, we only consider the case where $k = 1$.

For vector $v$ with $v_1 = 0$, we define $v' = (v_2, v_3, \cdots, v_p)^T$ and we can always identify a $(p-1) \times (p-1)$ orthogonal matrix $T'$ such that $T'v' = \|v'\|_2 e_1'$ where $e_1'$ is a $(p-1) \times 1$ unit vector with the first coordinate being 1. Now we define a new orthogonal matrix $T$ as

$$T = \begin{pmatrix} 1 & 0 \\ 0 & T' \end{pmatrix}$$

and we have

$$Tv = \begin{pmatrix} 1 & 0 \\ 0 & T' \end{pmatrix} \begin{pmatrix} 0 \\ v' \end{pmatrix} = \begin{pmatrix} 0 \\ \|v\|_2 e_1' \end{pmatrix} = \|v\|_2 e_2. \quad \text{and} \quad e_1^T T^T = e_1^T \begin{pmatrix} 1 & 0 \\ 0 & T'^T \end{pmatrix} = e_1^T$$

Therefore, we have

$$e_1^T H H^T v = e_1^T T^T T H H^T T^T T v = e_1^T T^T H H^T T^T e_2 = \|v\|_2 e_1^T \tilde{H} \tilde{H}^T e_2.$$

Since $H$ follows $MACG(\Sigma)$, $\tilde{H} = T^T H$ follows $MACG(T^T \Sigma T)$ for any fixed $T$. Therefore, we can apply Lemma 3 in Wang et al. (2015) or Lemma 5 again to obtain that

$$\mathbb{P}\left( |e_1^T X^T (XX^T)^{-1} X v| \geq \frac{\|v\|_2 c_4 c^* t}{c_*} \frac{\sqrt{n}}{p} \right) = \mathbb{P}\left( |e_1^T H H^T v| \geq \frac{\|v\|_2 c_4 c^* t}{c_*} \frac{\sqrt{n}}{p} \right)$$

$$= \mathbb{P}\left( \|v\|_2 |e_1^T \tilde{H} \tilde{H}^T e_2| \geq \frac{\|v\|_2 c_4 c^* t}{c_*} \frac{\sqrt{n}}{p} \right) = \mathbb{P}\left( |e_1^T \tilde{H} \tilde{H}^T e_2| \geq \frac{c_4 c^* t}{c_*} \frac{\sqrt{n}}{p} \right) \leq 5 e^{-Cn} + 2 e^{-t^2/2}.$$

Applying the above result to $v = (0, \beta_*^{(-1)})$ we have

$$\frac{1}{n} |\tilde{x}_1^T \tilde{X}^{(-1)} \beta_*^{(-1)}| = \frac{1}{n} |e_1 \tilde{X}^T \tilde{X} v| = \frac{1}{n} \left| e_1 X^T \left( \frac{XX^T}{p} \right)^{-1} X v \right| = \frac{p}{n} |e_1 X^T (XX^T)^{-1} X v| \leq \frac{c_4 c^* t}{c_*} \frac{\|\beta_*\|_2}{\sqrt{n}},$$

with probability at least $1 - 5 e^{-Cn} - 2 e^{-t^2/2}$.

In addition, we know that $\sigma_0^2 = var(Y) = \beta_*^T \Sigma \beta_* + \sigma^2$ and thus

$$\|\beta_*\|_2 \leq \sqrt{\frac{\sigma_0^2 - \sigma^2}{c_*}}.$$

Consequently, we have

$$\mathbb{P}\left( \frac{1}{n} |\tilde{x}_1^T \tilde{X}^{(-1)} \beta_*^{(-1)}| \geq \frac{\sqrt{\sigma_0^2 - \sigma^2} t}{\sqrt{n}} \right) \leq 2 \exp\left( -\frac{c_*^3}{2 c_4^2 c^{*2}} t^2 \right) + 5 e^{-Cn}.$$

Applying the result to any $k \in Q$ and taking the union bound gives the result in Lemma 6. $\quad \square$

## B.2: Proof of Theorem 1 and 2

We assemble all previous results to prove these two theorems.

**Proof of Theorem ?? and ??.** We just need to verify the Condition 1 and 3 listed in Theorem 0 and the variants of Condition 2 in two corollaries.

First, we verify Condition 1. Taking $M_1 = \frac{c_1 c_*}{c^*}$ and $M_2 = \frac{c_2 c^*}{c_*}$ and using Lemma 5, we have that

$$\mathbb{P}\left(M_1 \leq \frac{1}{n}|\tilde{x}_i^T \tilde{x}_i| \leq M_2, \ \forall i \in Q\right) \geq 1 - 4pe^{-Cn}.$$

Next, we verify Condition 3, which follows immediately from Lemma 6. For any $l \in \{1, 2, 3, \cdots, m\}$, we have

$$\max_l \frac{1}{n}\|\tilde{X}^{(l)}\tilde{W}^{(l)}\|_\infty \leq \max_{k \in Q} \frac{1}{n}|\tilde{x}_k^T \tilde{X}^{(-k)}\beta_*^{(-k)}| + \max_{k \in Q} \frac{1}{n}|\tilde{x}_k^T \tilde{\varepsilon}| \leq \frac{\sqrt{2}\sigma_0 t}{\sqrt{n}},$$

with probability at least $1 - 2p\exp\left(-\frac{c_* c_0^2}{2c^* c_2(1-c_0)^2}t^2\right) - 2p\exp\left(-\frac{c_*^3}{2c_4^2 c^{*2}}t^2\right) - 9pe^{-Cn}$. Taking $t = A\sqrt{\log p}/(2\sqrt{2})$ for any $A > 0$, we have

$$\mathbb{P}\left(\max_l \frac{1}{n}\|\tilde{X}^{(l)}\tilde{W}^{(l)}\|_\infty \geq \frac{1}{2}A\sigma_0\sqrt{\frac{\log p}{n}}\right) \leq 2p^{1-C_1 A^2} + 4p^{1-C_2 A^2} + 9pe^{-Cn},$$

where $C_1 = \frac{c_* c_0^2}{16c^* c_2(1-c_0)^2}$ and $C_2 = \frac{c_*^3}{16c_4^2 c^{*2}}$. This also indicates that $\lambda_n$ should be chosen as

$$\lambda_n = A\sigma_0\sqrt{\frac{\log p}{n}}.$$

Finally, we verify the two conditions in Corollary 1 and 2. Notice that Lemma 5 indicates that

$$\mathbb{P}\left(\max_{i \neq j} \frac{1}{n}|\tilde{x}_i^T \tilde{x}_j| \geq A\sqrt{\frac{\log p}{n}}\right) \leq 2p^{1-8C_2 A^2/c_*} + 5pe^{-Cn} \leq 2p^{1-C_2 A^2} + 5pe^{-Cn}.$$

Therefore, the two conditions in Corollary 1 and 2 will be satisfied as long as

$$A^2 \gamma_1^2 s^2 \frac{\log p}{n} \leq 1 \quad \text{and} \quad A^2 \gamma_1^2 R^2 \left(\frac{\log p}{n}\right)^{1-r} \leq 1.$$

Now we have verified that the three conditions hold for all subsets of the data. Let $\hat{\beta}^{(l)}$ and $\beta_*^{(l)}$ denote the estimate and true value of the coefficients on the $l^{th}$ worker and define $s_l = \|\beta_*^{(l)}\|_0$ and $R_l = \|\beta_*^{(l)}\|_r^r$. Applying Corollary 1 and 2 to each subset and taking $\gamma_1 = 64/M_1$ we have

$$\|\hat{\beta}^{(l)} - \beta_*^{(l)}\|_\infty \leq \frac{5A\sigma_0}{M_1}\sqrt{\frac{\log p}{n}} \quad \text{and} \quad \|\hat{\beta}^{(l)} - \beta_*^{(l)}\|_2^2 \leq \frac{72A^2\sigma_0^2}{M_1^2}\frac{s_l \log p}{n}$$

for $l = 1, 2, \cdots, m$ and $\beta_*$ being s-sparse. For $\beta_* \in \mathbb{B}(r, R)$, we have

$$\|\hat{\beta}^{(l)} - \beta_*^{(l)}\|_\infty \leq \frac{12A\sigma_0}{M_1}\sqrt{\frac{\log p}{n}} \quad \text{and} \quad \|\hat{\beta}^{(l)} - \beta_*^{(l)}\|_2^2 \leq \left(\frac{72}{M_1^2} + 38\right)(A\sigma_0)^{2-r}R_l\left(\frac{\log p}{n}\right)^{1-\frac{r}{2}}.$$

Notice that $\|\hat{\beta} - \beta_*\|_2^2 = \sum_{l=1}^m \|\hat{\beta}^{(l)} - \beta_*^{(l)}\|_2^2$, $s = \sum_{l=1}^m s_l$, $R_l = \sum_{l=1}^m R_l$. Taking summation over $l$

and replacing $M_1$ by $c_1 c_* / c^*$ completes the whole proof. $\qquad\square$