[Reviews · NeurIPS 2016]

Reviewer 1

Summary

The paper presents a method for regression in linear models in the domain of high dimensional problems, when the dimensionality far exceeds the sample size. The proposed framework involves decorrelating the features, splitting the data matrix columnwise, and then independently learning the parameters local to the sub-problem. The authors also provide consistency guarantees with respect to sign and various norms for the Lasso problem.

Qualitative Assessment

The paper deals with learning linear regression models in a distributed environment when the dimensionality exceeds the sample size. It is an important problem in machine learning and statistics. The theoretical contribution, towards proving the consistency of the estimators obtained by running the DECO algorithm, seems quite solid. Some of the points which are not clear, and may limit the potential impact of the work are : - The proved bounds are independent of m, which can possibly be a drawback from the point of practical implementation of DECO framework. For instance, given a fixed number of computing units, is there a good split of the number of features to be selected for each such unit? - It would be great to have comparisons with competitive approaches such as those given in http://www.math.ucla.edu/~wotaoyin/papers/GRock/ and LibLinear. These tools also seems to address similar problems and code/data publicly available. - On the empirical evaluation, the biggest dataset used (for electricity workload) has less than a billion entries which can be stored on a modern day desktop. In this regard, perhaps the paper has not fully explored the potential of the proposed DECO framework. Overall, on one hand, the consistency results as derived in the theoretical part of the paper seem nice. It would be great to strengthen the experimental section with more comparisons with existing approaches and bigger datasets for parallel training of sparse regression models.

Confidence in this Review

1-Less confident (might not have understood significant parts)


Reviewer 2

Summary

The paper presents a feature-wise partitioning approach for distributed sparse regression. Unfortunately, the results are rather incremental for the level of NIPS, as the result only holds for random design matrices, and the paper in its current form lacks discussion of several lines of related work and experimental baselines.

Qualitative Assessment

The paper presents a feature-wise partitioning approach for distributed sparse regression. Unfortunately, the results are rather incremental for the level of NIPS, as the result only holds for random design matrices, and the paper in its current form lacks discussion of several lines of related work and experimental baselines. While I do definitely like the conceptual idea of the partitioning followed by de-correlation, the presented theory falls short of expectations as it only holds for random design matrices. The paper however does not clearly explain the novelty and differences over [9]. Also, in addition to [9], relations to related work [B,C] are not sufficiently discussed in the current version. Almost matching upper and lower bounds for the amount of communication are in the meantime well-studied [C]. The approach presented here needs to be set into this relevant context as well. Clarity of presentation can be improved. The presented experiments are comprehensive in terms of datasets used, however not comprehensive in term of competing algorithms (many distributed solvers for L1 exist and the most prominent ones should be included in the comparison). MINOR: - abstract, line 6 "feature space partitioning is more effective when p >> n". This should be supported by a result. Might need to tune down, as most algorithms scale with p*n no matter which of the two possible partitionings is chosen. - conditional number -> condition number REFERENCES: [A] Arjevani, Y., & Shamir, O. (2015). Communication Complexity of Distributed Convex Learning and Optimization (pp. 1747–1755). NIPS 2015 - Advances in Neural Information Processing Systems 28. [B] Heinze, C., McWilliams, B., & Meinshausen, N. (2016). DUAL-LOCO: Distributing Statistical Estimation Using Random Projections (pp. 875–883). Presented at the AISTATS - Proceedings of the th International Conference on Artificial Intelligence and Statistics. [C] McWilliams, B., Heinze, C., Meinshausen, N., & Krummenacher, G. (2014). LOCO: Distributing Ridge Regression with Random Projections. arXiv UPDATE AFTER REBUTTAL: I thank the authors for clarifying the relation of the idea to [B,C]. I believe the authors can strengthen the paper for a future submission along the lines suggested.

Confidence in this Review

3-Expert (read the paper in detail, know the area, quite certain of my opinion)


Reviewer 3

Summary

The authors proposed a novel method, named DECO, for variable selection and parameter estimation in high-dimensional linear models. By partitioning the high-dimensional feature space into several parts and allocating them to distributed workers, this method substantially reduces the computational cost. Surprisingly, it also maintains desirable statistical properties. Numerical experiments are provided to back up the theory.

Qualitative Assessment

The proposed method in this paper is interesting and attractively both computationally and theoretically. Numerical experiments are convincing as well. In practice, how large m would the authors recommend to use?

Confidence in this Review

2-Confident (read it all; understood it all reasonably well)


Reviewer 4

Summary

This paper proposes the DECO algorithm for sparse regression which partitions the features of a high-dimensional data across multiple machines, decorrelates the data in each partition, solves a sparse regression sub-problem at each machine, and then combines the results. The authors prove performance guarantees and experimentally show performance improvements over Lasso variants.

Qualitative Assessment

This is an elegant, intuitive algorithm that to my knowledge has not appeared in previous literature. The technical contribution combines previous results in a straightforward but novel way. Distributed sparse regression is an important topic, and this result could potentially have a big impact. However, there is somewhat of a disconnect between the theoretical guarantees and experiments. Instead of running the same experiments on various dependency models for $\Sigma$, it would be more interesting to check the tightness of Theorems 1 and 2 on a smaller set of models by computing parameters used in the proofs (such as the condition number of $\Sigma$). Despite having a large Experiments section, several empirical results are left unexplained. In Table 1, the authors claim that when variables are independent, lasso-naive performs similariy to DECO-2. However, this also the case in Model (iii) when the predictors have a group dependency structure. In addition, the authors should explain why DECO-3 and lasso-refine were not run for model (v). In Section 5, performance of DECO-3 is averaged over randomness in partitioning. In distributed applications, it may be more important to consider worst-case performance and robustness to partitioning. Furthermore, it seems quite counterintuitive to manually take interaction terms (increaseing the dimensionality), only to immediately decorrelate as the first step of DECO. In Table 2, the authors should comment on why lasso-naive performs worse than Null when it is a strictly richer model. Organizationally, I would have preferred putting more of the proofs in the main text and moving some of the experiments to supplemental material. This is especially true because some models are only mentioned in Table 1 without further discussion. Also, "Section 5.1" is referenced on line 250 when it no longer appears in the manuscript.

Confidence in this Review

2-Confident (read it all; understood it all reasonably well)


Reviewer 5

Summary

This paper discusses a new distributed sparse regression method. In contrast with many methods, this work considers partitioning the features space, as opposed to the sample space. The proposed algorithm allows to decorrelate the distributed groups of features in a parallel way so that the distributed regression avoids features correlations. A theory is presented to analyze the algorithm. Both synthetic and real data are used to demonstrate the efficiency of the proposed approach.

Qualitative Assessment

The paper is well written and reads easily. The theory is gently exposed while the proofs are left to supplementary materials. The proposed method is demonstrated on a wide set of experiments. The exposition would benefit from a glimpse at the main ideas of the proof in the main text.

Confidence in this Review

2-Confident (read it all; understood it all reasonably well)


Reviewer 6

Summary

The basic idea is simple: decorrelate features to allow for embarrassingly parallel feature-space partitioning and optimization. The authors show that this precomputation step scales nicely in terms of number of partitions and dimensionality.

Qualitative Assessment

Model parallelism in practice is often stymied by feature correIations, so this method has potential to be very useful in the future. I didn't check the proofs, but the theoretical results are good and backed up by experiments. One drawback: most experiments were quite small-scale, with problem sizes where you'd probably not need to partition your data anyways. Only the electricity load dataset seemed to be of a realistic size. Also, are there settings where decorrelation runtime is so expensive so that this method isn't feasible?

Confidence in this Review

2-Confident (read it all; understood it all reasonably well)